# Immunogenicity, Safety, and Anti-Viral Efficacy of a Subunit SARS-CoV-2 Vaccine Candidate in Captive Black-Footed Ferrets (*Mustela nigripes*) and Their Susceptibility to Viral Challenge

**DOI:** 10.3390/v14102188

**Published:** 2022-10-04

**Authors:** Ariel E. Leon, Della Garelle, Airn Hartwig, Elizabeth A. Falendysz, Hon S. Ip, Julia S. Lankton, Tyler N. Tretten, Terry R. Spraker, Richard Bowen, Tonie E. Rocke

**Affiliations:** 1U.S. Geological Survey, National Wildlife Health Center, 6006 Schroeder Rd., Madison, WI 53711, USA; 2U.S. Fish and Wildlife Service, National Black-Footed Ferret Conservation Center, 19180 North East Frontage Road, Carr, CO 80612, USA; 3Department of Biomedical Sciences, Colorado State University, 3107 Rampart Road, Fort Collins, CO 80523, USA; 4Department of Microbiology, Immunology and Pathology, Colorado State University, 2450 Gillette Dr, Fort Collins, CO 80526, USA

**Keywords:** SARS-CoV-2, black-footed ferrets, mustelids, vaccination, experimental challenge

## Abstract

A preliminary vaccination trial against the emergent pathogen, SARS-CoV-2, was completed in captive black-footed ferrets (*Mustela nigripes;* BFF) to assess safety, immunogenicity, and anti-viral efficacy. Vaccination and boosting of 15 BFF with purified SARS-CoV-2 S1 subunit protein produced a nearly 150-fold increase in mean antibody titers compared to pre-vaccination titers. Serum antibody responses were highest in young animals, but in all vaccinees, antibody response declined rapidly. Anti-viral activity from vaccinated and unvaccinated BFF was determined in vitro, as well as in vivo with a passive serum transfer study in mice. Transgenic mice that received BFF serum transfers and were subsequently challenged with SARS-CoV-2 had lung viral loads that negatively correlated (*p* < 0.05) with the BFF serum titer received. Lastly, an experimental challenge study in a small group of BFF was completed to test susceptibility to SARS-CoV-2. Despite viral replication and shedding in the upper respiratory tract for up to 7 days post-challenge, no clinical disease was observed in either vaccinated or naive animals. The lack of morbidity or mortality observed indicates SARS-CoV-2 is unlikely to affect wild BFF populations, but infected captive animals pose a potential risk, albeit low, for humans and other animals.

## 1. Introduction

The novel coronavirus SARS-CoV-2, the cause of the COVID-19 pandemic, was initially considered a potential threat to black-footed ferrets (*Mustela nigripes*, BFF), one of the most endangered mammals in North America. To augment their populations and enhance their recovery in the wild, BFF are held and bred at several facilities in North America. In the spring and summer of 2020, SARS-CoV-2 virus infected and caused severe disease, including respiratory distress and death, in related captive mustelids—two species of farm-raised mink (*Mustela lutreola* and *Neovison vison*). Outbreaks on mink farms led to depopulation at several facilities in the Netherlands [1] and declines at farms in the United States [2]. Moreover, reciprocal transmission among humans and mink has been documented [3], demonstrating the potential risk infected caretakers may pose for captive animals and vice versa [4]. The domestic ferret (*Mustela putorius furo)*, another related mustelid species, is also known to be susceptible to SARS-CoV-1 [5] and SARS-CoV-2 [6,7] and has been used in experimental infection studies. Transmission of SARS-CoV-2 among domestic ferrets was shown to occur via direct contact and air-borne respiratory droplets [7]. Given the genetic similarities between mink, domestic ferrets, and BFF, it was considered likely that BFF were also susceptible to infection by SARS-CoV-2. Although no evidence of natural infection from the virus has been detected in BFF to date, SARS-CoV-2 infection has occurred in several captive wildlife species, including lions and tigers at the Bronx Zoo [8], gorillas at the San Diego Zoo [9], and numerous other species, including snow leopard, otter, and spotted hyena [10].

In 1981, a small number of BFF, the last known extant population in the world, was discovered near Meeteetse, Wyoming [11]. When locally circulating disease agents (canine distemper virus and *Yersinia pestis*, the causative agent of plague) reduced this population, the last remaining BFF were captured and brought into captivity to establish a breeding population and recovery program. Today, the U.S. Fish and Wildlife Service’s (USFWS) National Black-Footed Ferret Conservation Center (NBFFCC) is one of six managed care and captive breeding facilities that work to ensure survival of this endangered species through the Association of Zoo and Aquarium’s Species Survival Plan (SSP). The NBFFCC houses approximately two-thirds of the world’s captive population of BFF, with the other third distributed among the Smithsonian Conservation Biology Institute, Louisville Zoo, Phoenix Zoo, Cheyenne Mountain Zoo, and the Toronto Zoo. The recovery program has been highly successful re-introducing BFF to 32 sites, 19 of which remain active today (Pete Gober, USFWS, oral communication, 2021).

Shortly after SARS-CoV-2 was detected in the United States in 2020, personnel at facilities housing BFF instituted protocols to guard animals against viral exposure, including the use of enhanced personal protective equipment (PPE), enhanced physical separation of ferrets, ultraviolet air filtration, subdivision of animals into smaller groups with designated caretakers, and other precautions. However, these protocols are expensive, interfere with the breeding program, and are difficult to maintain for long periods of time.

As an alternative, and to provide additional protection to the animals, a pilot study was proposed to assess whether immunization of BFF using commercially available viral proteins could elicit a protective immune response against the virus. Previous studies in mice and hamsters with SARS-CoV-1 demonstrated that the CoV spike protein, particularly the receptor binding domain, is immunogenic and can successfully protect immunized animals against challenge with the virus [12,13]. If successful, a SARS-CoV-2 vaccine could be used as part of a larger effort to reduce the risk to BFF of contracting COVID-19. Vaccination of BFF could also reduce the risk of SARS-CoV-2 transmission from infected animals to human caretakers, should the virus be introduced into these facilities. Our specific objectives were to assess whether immunization of BFF is safe, elicits a sustained humoral immune response, and whether elicited antibodies were protective in vivo using a passive transfer and challenge study in transgenic mice, expressing the hACE-2 receptor. Following the early findings of the pilot vaccination study, additional BFF at the NBFFCC were vaccinated, and a challenge trial was conducted on post-reproductive BFF to assess their susceptibility to SARS-CoV-2.

## 2. Materials and Methods

### 2.1. Immunization of Black-Footed Ferrets

A group of 24 BFF (all males, from 1 to 4 years old), housed in separate cages, were selected for the study. The selected animals were housed at NBFFCC in a separate building from the remaining population for the duration of the study. Fecal samples were collected from all BFF prior to vaccination and tested via polymerase chain reaction (PCR) for ferret enteric coronavirus infection (IDEXX Laboratories, Inc, Westbrook, Maine USA). At the U.S. Geological Survey (USGS) National Wildlife Health Center (NWHC), the SARS-CoV-2 S1 subunit protein, engineered from sequences from the original strain of the virus (2019-nCoV/USA-WA1/2020; Leinco Technologies, Inc, St. Louis, MO, USA) was mixed with an adjuvant (alum) overnight and then injected into 15 BFF (via subcutaneous injection) in 50 µg doses in a total volume of 0.5 mL. The dose selected was that used in the successful immunization of Syrian hamsters against SARS-CoV-1 [13]. Control animals (*n* = 9) were given a sham vaccination of diluent with alum only. Three weeks later, a second vaccination (boost) or sham injection was given similarly to treated animals and controls, respectively (Figure 1). The BFF were observed daily after vaccination for any signs of morbidity.

### 2.2. Serology

Black-footed ferrets were anesthetized, and blood was drawn from either the jugular or cranial vena cava at the time of initial vaccination, at the time of the boost (3 weeks later), and at 2–3 weeks and 12 weeks post-boost (Figure 1). Sera were shipped to NWHC to assess antibody titers to the S1 protein. A direct enzyme linked immunosorbent assay (ELISA) using horseradish peroxidase labelled anti-ferret IgG was optimized for the detection of BFF anti-SARS-CoV-2 antibodies. Briefly, ELISA plates were coated with 3 mg/mL of protein (the same protein used to immunize the ferrets) in a volume of 50 ul/well. Serum samples were diluted 4-fold starting at 1:160. The highest dilution that was positive (defined as exceeding the mean of 4 negative control samples by 3 standard deviations) was considered the end point, and its reciprocal value was recorded as the titer (Table 1). Titers < 1:160 were recorded as 1:40 for analyses, and titers of 1:160 or less were considered background. Antibody titers were charted over time and compared to the matched pre-vaccination titer for each animal. Serum samples from BFF collected 2–3 weeks post-boost were also tested for in vitro virus neutralization activity using plaque reduction neutralization assays against the strain of SARS-CoV-2 isolated from the first U.S. patient [14]. All samples were screened at a dilution of 1:8, any samples with less than 50% neutralization were reported as negative at this stage, and any positives were tested at further dilutions, with a maximum dilution of 1:256 (Table 1). 

### 2.3. In Vivo Passive Serum Transfer Studies

To determine if BFF antibodies elicited by vaccination were protective against SARS-CoV-2, a passive serum transfer study was conducted in transgenic mice expressing human ACE-2 receptors (K18-hACE2 mice, Jackson Laboratory, Bar Harbor ME, USA 04609). After 2 days of acclimation, sera from vaccinated and control BFF collected 2–3 weeks and 12 weeks post-boost were transferred intraperitoneally (IP) into each mouse (*n* = 48) in volumes of 0.5 mL (Figure 1). For serum samples with less than 0.5 mL volumes, saline was added to ensure all animals received a 0.5 mL transfer. Four additional mice received 20 ug monoclonal anti-S1 antibodies (Leinco Technologies, Inc, St. Louis, MO, USA) in the same volume, 500 ul, and were used as positive controls. Twenty-four hours post-transfer, the mice were inoculated intranasally with approximately 10^4^ tissue culture infectious dose (TCID)_50_ of SARS-CoV-2 (20 µL per nostril; vero passage two of 2019-nCoV/USA-WA1/2020 originally provided by BEI Resources as NR-52281). An additional four mice (not treated with antibody) received a sham inoculation and served as negative controls for tissue assays. The mice were weighed daily, and signs of morbidity, such as labored breathing and disinclination to move, were recorded. Six days after inoculation, all the mice were euthanized to collect lung and nasal turbinate tissue for viral quantification via reverse-transcription (RT)-PCR and TCID_50_ determination. Tissue samples from brain, lungs, nasal turbinates, small and large intestines, spleen, and liver were also collected for histology to assess pathology related to SARS-CoV-2 infection [15], although only results from the brain and lungs are reported here. Viral titers in lungs were compared between mice that received serum from vaccinated BFF, those that received serum from BFF controls, and negative controls that were not infected. 

### 2.4. Histopathology of Mice from Serum Transfer Study

Tissues were collected in 10% neutral buffered formalin at necropsy, processed for light microscopy, and stained with hematoxylin and eosin according to standard protocols [16]. The brain and lungs were assessed by light microscopy for evidence of SARS-CoV-2 infection (e.g., [15]). In the brain, parameters assessed included hemorrhage (presence/absence and number of sites in most affected section), perivascular inflammation (presence/absence and number of sites in most affected section), and gliosis (presence/absence). In the lungs, parameters assessed included hemorrhage (presence/absence), perivascular inflammation (presence/absence), interstitial pneumonia (presence/absence and percentage of tissue affected expressed as a range), and edema (presence/absence).

Perivascular inflammation in the brain and interstitial pneumonia in the lung were graded according to the following scales. Number of vessels with perivascular inflammation in the brain: absent (grade 0); 1–10 (grade 1); 11–20 (grade 2); over 20 (grade 3). Percentage of most affected lung lobe with interstitial pneumonia: absent (grade 0); less than 10% (grade 1); 10–20% (grade 2); >20–40% (grade 3); >40–50% (grade 4); over 50% (grade 5). A total grade for each animal was assigned by adding the brain and lung grades. 

### 2.5. Challenge of Vaccinated and Unvaccinated BFF with SARS-CoV-2 Virus

Six post-reproductive male and female BFF, 4 to 7 years of age, were anesthetized for physical examinations, collection of blood, and subcutaneous implantation of temperature transponders (Destron Fearing LifeChip with Bio-Thermo Technology, Dallas-Fort Worth, TX, USA). Two of these ferrets had been previously vaccinated with SARS-CoV-2 protein, one as part of the experimental trial described above (Methods 2.1) and one thereafter (Methods 2.2). All animals were then transferred to a BSL3 containment facility at Colorado State University and acclimatized for three days prior to challenge. They were housed individually and fed the standard supplemented raw horsemeat diet recommended by the BFF SSP (Milliken Meat Products, Ltd., Markham, ON, Canada).

The six BFF were lightly anesthetized with a mixture of ketamine and xylazine, weighed, and challenged by intranasal instillation of 0.5 mL containing 10^5^ plaque-forming units of the SARS-CoV-2 virus (Vero passage two of 2019-nCoV/USA-WA1/2020 originally provided by BEI Resources as NR-52281). Clinical observations and body temperatures from the transponders were recorded daily. On days 1, 3, 5, and 7 following virus challenge, the ferrets were anesthetized as before, weighed, and oral swabs and nasal flushes collected for virus titration. Two of the animals were euthanized three days post-challenge to collect tissues for virus titration and histology. The remaining four BFF were euthanized and necropsied 14 days post-challenge. Virus titers in swabs and nasal flush fluid were determined by plaque assay on Vero cells and neutralizing antibody titers by plaque reduction neutralization [17]. Hematoxylin and eosin-stained sections of ferret tissues were examined for histopathologic lesions. Adjacent sections were evaluated for viral antigen by immunohistochemical staining using a rabbit antibody to SARS-CoV nucleocapsid protein (Novus Biologics, Littleton, CO, USA), with positive and negative controls from infected and non-infected hamsters.

### 2.6. Analyses

Data were analyzed using R version 4.1.1 (R Core Team 2021 [18]). For responses to vaccination, BFF serum ELISA titers post-vaccination (Table 1—pre-boost, 2–3 weeks and 12 weeks post-boost) were averaged by age of the BFF (Table 1—age (years)). A simple linear regression was used to determine if age at vaccination and week post-boost significantly predicted serum titer. Age was treated as a continuous variable, while week post-boost was treated as a categorical factor, including three time points—pre-boost, 2–3 weeks, and 12 weeks post-boost. 

To determine how treatment affected viral loads in the lungs of SARS-CoV-2 challenged mice, a linear model (excluding the 4 negative control mice) was used. Treatment was treated as a categorical variable including monoclonal antibody, positive serum—2 or 3 weeks post-boost, positive serum—12 week post-boost, and negative serum. A generalized linear model (GLM) assuming a gamma distribution with an inverse link function (lme4 package in R, [19]) was used to assess how log_10_ viral loads among mice that received a serum transfer (positive serum and negative serum) were influenced by the titer of the serum received. In addition, for animals that received a serum transfer, the BFF serum volume included in their IP injection was analyzed as a continuous independent variable in a GLM with a gamma distribution to determine the effect on log_10_ viral loads. Certain serum samples had less than 0.5 mL volumes available for injecting mice, but saline was added to ensure all animals received a 0.5 mL inoculation. Lastly, a GLM, again assuming a gamma distribution, was used to assess how lung grade predicted viral loads in the lung (excluding the 4 control animals); lung grade was treated as a continuous predictor variable for log_10_ viral copies. A gamma distribution was used in these analyses because they did not meet assumptions of normality and log_10_ viral copies in the lung is a positively skewed continuous variable. For all models, overall variable level effects were analyzed using a Type III Likelihood Ratio test using the car package in R [20].

## 3. Results

### 3.1. Ferret Immunization

A total of 3 of the 24 ferrets included in this study were PCR positive (with no clinical symptoms) for ferret enteric coronavirus prior to vaccination, but this finding was not correlated to pre- or post-vaccination antibody titers, so all ferrets were retained in the analysis. Upon immunization with SARS-CoV-2 purified protein, the only adverse effect noted was the formation of small hard spherical subcutaneous nodules at the injection site in some individuals. The nodules did not cause any irritation or apparent discomfort for the animals and dissipated after several days. Weights, appetites, feces, and activity level all remained stable throughout the study period. No respiratory or other clinical signs were observed in any ferrets.

### 3.2. Serology

Sera collected prior to vaccination, prior to boost, and after boost (2–3 weeks and 12 weeks post-boost) were measured by ELISA (Table 1). Prior to vaccination, antibody titers were ≤1:160 for all animals, and no change in titer was noted for controls post-vaccination. In contrast, for vaccinated BFF, mean antibody titers increased 60-fold after the first vaccine injection, with two animals not responding (#8847 and #9598). Following the boost, mean titers increased to 150-fold that of pre-vaccination levels within 2–3 weeks, but by 12 weeks, titers had declined in nearly all animals, with mean titer only 23-fold that of pre-vaccination levels (Figure 2 and Figure 3). A negative relationship was detected between the age of inoculated BFF and mean serum ELISA titers post-vaccination, with the youngest animals responding the greatest to vaccination (Figure 3). Both age at vaccination and week post-boost significantly predicted ELISA serum titers (Linear model: Pre-boost (intercept): 10090 ± 2279; 2–3-weeks post-boost: 7166 ± 1973; 12 weeks post-boost: −1863 ± 1973; Age: −2252 ± 720.7; R^2^ = 0.805, Week Post-Boost: F (2, 8) = 11.67, *p*= 0.004; Age: F (1, 8) = 9.76, *p*= 0.014; Figure 2 and Figure 3).

Plaque reduction neutralization (PRNT_50_) assays were also conducted to assess levels of viral neutralizing activity in the serum at 2–3 weeks post-boost. Although end points were not determined for all animals, 11/15 were positive, ranging in titer from 1:8 to 1:256 (Table 1), but PRNT titers only moderately correlated with ELISA titers (Pearson’s correlation: 0.5314; Appendix A). The ELISA titers better predicted results in the transgenic mouse serum transfer study compared to the PRNT results (data not shown). As such, only the ELISA results were used for further analyses. 

### 3.3. Passive Serum Transfer Studies in Mice

Sera collected at 2–3 weeks and 12 weeks post-boost from 15 vaccinated BFF and 9 that received a placebo were inoculated IP into K18-hACE2 mice. Within a few hours after serum inoculation and prior to challenge with SARS-CoV-2 on the same day, it was noted that many of the mice that received ferret serum were unexpectedly ill, with severe signs of discomfort and disinclination to move, eat, or drink. Four mice that received monoclonal anti-S1 antibodies did not show signs of illness, nor did four negative control animals (Appendix A). Ultimately, 13 mice (5 that received serum from vaccinated BFF and 8 that received serum from negative controls) were euthanized prior to challenge based on pre-determined humane end points. The remaining animals were given subcutaneous fluids (0.5 mL 0.9% NaCl; ICU Medical, San Clemente, CA, USA) and allowed to recover overnight. The following morning, all mice appeared apparently healthy and were challenged with SARS-CoV-2 as planned. Ultimately, 35 mice that received ferret serum and 4 mice that received monoclonal antibodies were challenged with SARS-CoV-2. On day 5 post-inoculation, some mice had lost >1 g body weight and some displayed lethargy, hunched back, and/or squinted eyes (Appendix A). The mice were euthanized as planned by CO_2_ asphyxiation on day 6 post-inoculation. Upon euthanasia and necropsy, tissues were collected for viral load and histologic examination. 

### 3.4. Viral Loads by Treatment and by ELISA Antibody Titers in Mice

A significant effect of treatment on viral loads quantified in the right lung of all mice at the end point of the study was detected. Mice that received anti-S1 monoclonal antibodies (AB) had significantly lower viral loads compared to the animals that received a serum transfer from either vaccinated (positive serum, 12 weeks post-boost) or control BFF (negative serum) but not lower than vaccinated animals that received positive serum at 2–3 weeks post-boost (Linear model: Monoclonal AB (intercept): 3.19 ± 0.738; Positive serum—2 or 3 weeks post-boost: 1.03 ± 0.873, *p* = 0.24; Positive serum—12 weeks post-boost: 3.28 ± 0.831, *p* < 0.001; Negative serum: 3.11 ± 0.874, *p* = 0.001; R^2^ = 0.432, F (3, 35) = 8.89, *p* < 0.001; Figure 4 and Figure 5). Mice that received serum from vaccinated BFF at 2–3 weeks post-boost had significantly lower viral loads than mice that received serum from unvaccinated controls (t = 3.14, *p* = 0.0034) as well as from vaccinated BFF at 12 weeks post-boost (t = 3.73, *p* < 0.001). Upon further investigation, reduced viral loads among serum-treated animals was significantly predicted by the BFF serum titers levels (GLM assuming a gamma distribution: (intercept): 1.49 × 10^−1^ ± 1.30 × 10^−2^; Serum Titer: 8.170 × 10^−6^ ± 2.314 × 10^−6^; LR = 18.5, df = 1, *p* < 0.0001; Figure 5). Moreover, serum volume given IP to mice significantly affected viral titers, with increased serum volume reducing viral titers in the lung (GLM assuming a gamma distribution: (intercept): 0.063 ± 0.039; IP Serum Volume: 0.00036 ± 0.00011; LR = 9.54, df = 1, *p* = 0.002; Appendix A). This result was driven by the 12-week post-boost serum samples. No effect of injection volume was seen in negative (from BFF given a placebo) serum injections, and 2–3 weeks post-boost samples were all a single volume, precluding an analysis of serum volume (Appendix A).

### 3.5. Histologic Examination of Tissues from Mice

Brain inflammation (lymphoplasmacytic perivascular cuffing; Appendix A) was only seen in the positive and negative serum groups (6/25 and 3/10, respectively) and is attributed to SARS-CoV-2 infection (Appendix A; Appendix A). The mean number of blood vessels with perivascular inflammation per section of brain examined was greater in the positive than in the negative serum group (12.7 versus 6). Similarly, gliosis in the brain (Appendix A) was only seen within the positive and negative serum groups (5/25 and 2/10, respectively) and is also attributed to SARS-CoV-2 infection. Hemorrhage within the brain was present within all treatment groups (13/25, 4/10, 1/4, and 4/4 for positive serum, negative serum, monoclonal antibody, and PBS negative control groups, respectively) and is considered most likely an artifact of euthanasia or necropsy.

In the lung, perivascular inflammation (Appendix A) was common in the positive serum, negative serum, and monoclonal antibody groups (21/25, 9/10, and 3/4, respectively) and was absent in the PBS group. Interstitial pneumonia (Appendix A), defined as expansion of alveolar septa and filling of alveoli by lymphocytes, plasma cells, histiocytes, and/or neutrophils, with type 2 pneumocyte hyperplasia, followed a similar pattern (20/25, 9/10, 4/4, and 0/4 mice for positive serum, negative serum, monoclonal antibody, and PBS groups, respectively). Both perivascular inflammation and interstitial pneumonia are attributed to SARS-CoV-2 infection. In comparison to the serum groups, the percentage of lung affected by areas of interstitial pneumonia was consistently low (less than 10%) in the monoclonal antibody group, resulting in a lower lung and overall grade and indicating the likelihood of less severe disease, despite a similar prevalence. Pulmonary edema was variably present in 6/25, 5/10, and 1/4 mice, in the positive serum, negative serum, and monoclonal antibody groups, respectively, and is considered likely associated with SARS-CoV-2 infection although less characteristic of infection than inflammatory lesions. Pulmonary edema was not noted in the PBS negative control group (0/4). Hemorrhage was present within the lung in almost all animals (24/25, 9/10, 4/4, and 4/4 mice for positive serum, negative serum, monoclonal antibody, and PBS groups, respectively) and, like hemorrhage within the brain, is considered most likely an artifact of euthanasia or necropsy.

In comparison to lung CDC N2 PCR results, the mean lung and total grades for mice with Ct <20 was three and three, respectively, while the mean lung and total grade for mice with Ct >35 was one and one, respectively, indicating correlation between viral load and severity of inflammation (Appendix A). Upon further analysis, lung grade significantly positively predicted viral load (GLM assuming a gamma distribution: (intercept): 0.215 ± 0.0177; Lung Grade: −0.0154 ± 0.00600; LR = 6.311, df = 1, *p* = 0.012). This association seems to be primarily driven by individuals that received positive BFF serum, where a positive correlation exists between viral load and lung grade, which is absent for both the negative serum and monoclonal antibody groups (Appendix A). Due to a lack of statistical power, we were unable to test an interactive term between lung grade and treatment type.

### 3.6. SARS-CoV-2 Challenge in Black-Footed Ferrets

None of the BFF displayed clinical signs of disease following virus challenge, and no changes in activity, behavior, or consumption of food were noted. Virus shedding, as determined by the presence of virus in oropharyngeal swabs or nasal flushing, was detected in five of the six BFF (Table 2). Shedding was detected through day 7 in three of the four animals followed until that time point. The single ferret that did not shed detectable quantities of virus (#8847) had been vaccinated, and its 90% reduction PRNT (PRNT_90_) titer 3 days prior to challenge was 320, which rose dramatically by 14 days post-challenge (Table 2 & Table 3). The antibody titers of the other vaccinated ferret (#8305) and the four unvaccinated ferrets were <10 at the time of challenge and rose modestly by day 14 post-challenge (Table 3). Virus was detected in turbinates of both ferrets euthanized 3 days post-challenge and in caudal lung of one of those ferrets, indicating robust virus replication in the upper respiratory tract (Table 2 & Table 3).

### 3.7. Histopathology of Challenged Black-Footed Ferrets

Histopathologic evaluation of tissues collected from ferrets euthanized on days 3 and 14 following virus inoculation revealed several common chronic background lesions unrelated to SARS-CoV-2 infection, likely due to advanced age. These lesions included moderate hepatic hydropic degeneration or accumulation of glycogen and fat (all BFF), moderate Ito cell hyperplasia in the liver, mild renal glomerulosclerosis and fibrosis (five of six BFF), and mild myocardial necrosis and fibrosis (two of six). Four of the six BFF had mild, focal patches of interstitial pneumonia that is consistent with SARS-CoV-2 infection in other animals, and three ferrets had mild to moderate suppurative rhinitis with visible small accumulations of Gram-negative bacteria (Appendix A). However, these lesions in the upper and lower respiratory tract were negative for viral antigens by immunohistochemistry and may have reflected pre-existing subclinical conditions or perhaps healing phases of previous non-SARS-CoV-2 viral infection.

## 4. Discussion

In the spring of 2020, little was known regarding the potential effect of SARS-CoV-2 in most wildlife species, whether captive or free-ranging. However, reports of illness and death in farm-raised mink, transmission between mink and their human caretakers [3], and the need for depopulation on some of these farms [1] were alarming prospects for managers of captive mustelids, including BFF. Even though a commercially licensed vaccine against SARS-CoV-2 was not yet available for mustelids or other animals, vaccination was considered the most feasible long-term strategy for protecting BFF against potential disease and preventing transmission of the virus from humans to ferrets and back again. To act proactively in this regard, USFWS authorized a preliminary vaccination trial in a small group of BFF raised in captivity to assess safety, immunogenicity, and anti-viral efficacy, as well as an experimental challenge study in a small group of post-reproductive animals. 

Upon vaccination and boosting of 15 BFF (3 weeks apart) with purified preparations of the SARS-CoV-2 S1 subunit protein at a single dosage (50 ug) mixed with alum, we demonstrated a significant rise in mean antibody titers against the virus, about 150-fold that of pre-vaccination titers. Antibody response was correlated with age, with younger animals (especially 1 year old) responding better than older animals (up to 4 years old), as previously seen in other systems [21], but in all vaccinated animals, antibody response declined rapidly with measured titers about 28-fold higher than pre-vaccination titers at 12 weeks post-boost. 

Anti-viral activity in the sera from vaccinated and unvaccinated BFF was determined both in vitro and in vivo. Plaque reduction neutralization assays on serum collected 2–3 weeks post-boost indicated 11 of 15 vaccinated animals had some virus neutralizing activity (from 1:8 to 1:256), but these results only moderately correlated to ELISA titers (Table 1). 

Serum transfer and challenge studies in K18-hACE2 mice were somewhat compromised by the unexpected reaction of the mice to ferret serum. We believe that the adverse reaction was a result of a hypersensitivity to an antigen present in the ferret serum. The four mice that received monoclonal antibodies did not show evidence of post-injection reaction. Other investigators have used similar methods to conduct passive transfer studies in K18-hACE2 mice using serum from non-human primates without adverse effects (Jorge Osorio, University of Wisconsin, Madison, oral communication, May 2022). Zheng et al. showed that intravenous infusion of convalescent plasma from humans in this mouse model did not cause negative effects and offered protection from severe SARS-CoV-2 disease [22]. Cytokine Release Syndrome (CRS) may have been the cause of the acute reaction and has been documented in mice and humans in response to monoclonal antibodies and other immunotherapies as well as SARS-CoV-2 infection [23]. In this case, the group receiving monoclonal antibodies did not show acute disease, while mice receiving serum from either vaccinated (with SARS-CoV-2 antibodies) and unvaccinated ferrets (without SARS-CoV-2 antibodies) did, supporting the hypothesis that the reactions were caused by ferret serum specifically. 

Despite the loss in sample size due to the reactions to serum transfer, our study still provided enough power to make statistical inferences. Although mean viral loads were not significantly different in mice that received serum from vaccinated or unvaccinated ferrets overall, a significant difference was evident among mice that received BFF serum collected 2–3 weeks post-boost and those that received serum from unvaccinated animals (Figure 4). It should be noted that serum samples collected from BFF 12 weeks post-boost had variable volumes available for transfer. The volume of serum transferred into a mouse significantly predicted viral loads in mice, where increased volumes produced lower viral loads in the lungs post-challenge (Appendix A). This suggests that higher volume availability of these samples for serum transfer may have produced a stronger protective effect in vivo than was recorded in this study. A significant negative correlation was also evident between viral load and ELISA antibody titers within the vaccinated group, where viral load decreased with increasing titer (Figure 5). Taken together, these results indicate vaccination of BFF can elicit a protective anti-viral response in vivo. 

To test susceptibility of BFF to SARS-CoV-2, as well as to begin to assess protective effects of vaccination in BFF, a small group of vaccinated (*n* = 2) and unvaccinated BFF (*n* = 4) was experimentally inoculated with SARS-CoV-2 virus. Unlike mink, and even with significant virus shedding and replication of SARS-CoV-2 in the upper respiratory tract for up to 7 days post-challenge, the challenged BFF, despite their advanced age and existing co-morbidities, did not develop any indication of clinical disease or significant pathologic lesions attributable to SARS-CoV-2 infection. These results are similar to outcomes reported in domestic ferrets, which showed signs of elevated body temperatures and viral shedding for up to 8 days post-challenge but also no mortality [5; Bowen et al. unpublished observations]. The lack of clinical disease, coupled with the species’ solitary lifestyle and demonstration that black-tailed prairie dogs (*Cynomys ludovicianus*), the primary prey of BFF, are not susceptible to SARS-CoV-2 infection [17], provides evidence that SARS-CoV-2 infection is unlikely to affect wild populations of this endangered species. 

Based on positive serologic and safety results of the vaccine trial and prior to the challenge study, USFWS personnel vaccinated 60% of the captive ferrets at NBFFCC in 2020. Even though challenge results indicate that BFF are not susceptible to severe disease or illness from SARS-CoV-2 infection, they can be experimentally infected and shed virus, which may pose a risk of viral transmission between them and human caretakers, animals in zoo settings, and other species when ex situ BFF are released into the wild. This risk is mitigated at NBFFCC through safety protocols; wearing PPE (an N95 respirator at minimum) is required of all those in the same room with a BFF and rarely is a ferret within 6 feet of a caretaker for more than 15 min per day. One of the two vaccinated BFF that were challenged with SARS-CoV-2 did not shed any virus after exposure. This male, 4 years old at vaccination and 5 at challenge, was the only animal with detectable neutralizing antibody against the virus prior to challenge; the other vaccinated animal (7-year-old female) had no detectable antibody at challenge and the highest rates of viral shedding and viral loads in tissues (Table 2 and Table 3). Because age significantly predicted antibody titer in our initial immunization trial, presumably vaccination of younger animals would result in more positive outcomes upon SARS-CoV-2 exposure, with little to no virus shedding, as observed in the challenged male that was vaccinated at a younger age (4 years). However, the rapid decline in antibody titers in BFF after vaccination indicates that their humoral response is transient or higher dosages of antigen are required. Similar rapid declines within 3–4 months of vaccination have been observed in humans after two doses of mRNA vaccines [24,25]. Experimental subunit vaccines have now become available for use in mink and other animals [26] that elicit higher antibodies in cats and dogs but results in mustelids are not yet available. In the future, a commercial vaccine may be used to vaccinate and boost susceptible captive animals of various species.

## Figures and Tables

**Figure 1 viruses-14-02188-f001:**
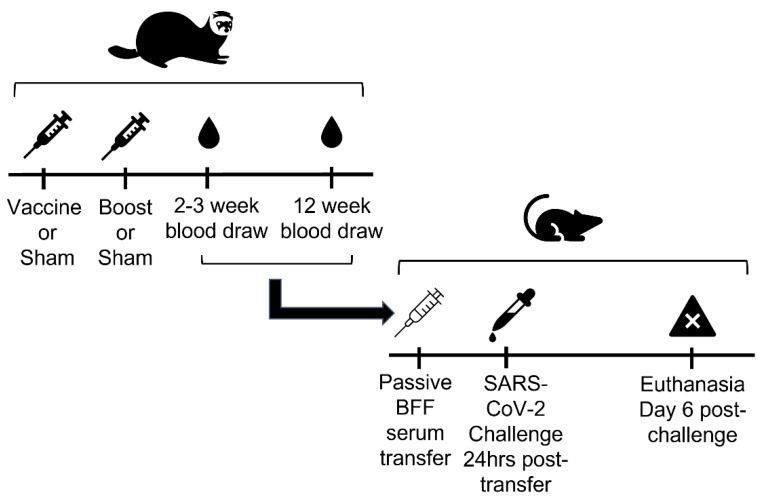
Experimental design and timeline of the immunization of black-footed ferrets and subsequent passive transfer study conducted in transgenic mice with serum from BFF. BFF were vaccinated with SARS-CoV-2 S1 subunit protein or given a sham inoculation, then a subsequent boost of the same vaccination or sham. They were bled 2–3 weeks and 12 weeks post-boost. This serum was used for a passive serum transfer into transgenic mice. Mice were subsequently challenged with SARS-CoV-2 to determine the protective effect of the BFF serum. The mice were euthanized 6 days post-inoculation.

**Figure 2 viruses-14-02188-f002:**
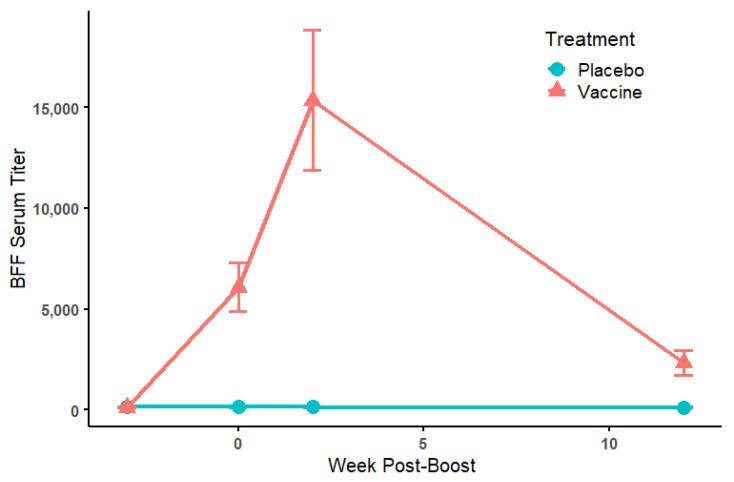
Mean black-footed ferret (BFF) anti-SARS-CoV-2 serum titers as measured by enzyme-linked immunosorbent assays (ELISA) by week post-boost. Error bars represent the standard error of the mean. BFF were vaccinated at week −3 and 0, with titers increasing after both prime and boost vaccinations and peaking at 2–3 weeks post-boost. No change in titer was observed in control animals receiving a placebo (blue line; circles). For the purposes of visualization, animals that were bled at 2 or 3 weeks post-boost were grouped here as a single data point.

**Figure 3 viruses-14-02188-f003:**
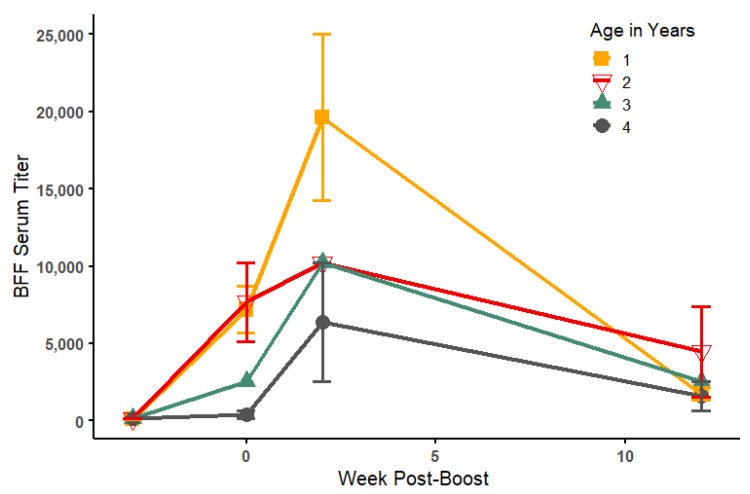
Mean anti-SARS-CoV-2 serum titers for immunized black-footed ferrets as measured by enzyme-linked immunosorbent assays (ELISA) by age and week post-boost. Error bars represent the standard error of the mean. For the purposes of visualization, animals that were bled at 2 or 3 weeks post-boost were grouped here as a single data point.

**Figure 4 viruses-14-02188-f004:**
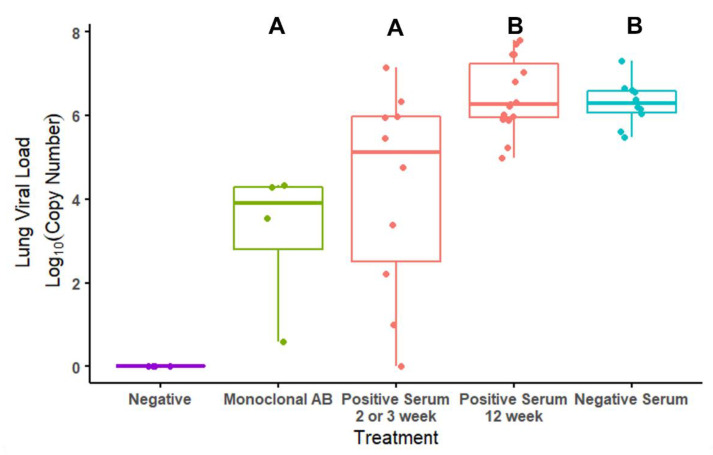
Quantitative log_10_ viral loads from the right lung of mice challenged with SARS-CoV-2 by treatment. Treatment indicates whether mice received a serum transfer from vaccinated (positive serum—either 2–3 weeks or 12 weeks post-boost; shown in red) BFF, unvaccinated (negative serum; shown in blue), or a monoclonal anti-S1 antibody (monoclonal AB; shown in green) prior to challenge. Negative treatment represents a negative control (shown in purple). These mice were given a placebo challenge. Each point represents an individual mouse. Different letters represent which groups significantly differ from each other (*p* ≤ 0.05).

**Figure 5 viruses-14-02188-f005:**
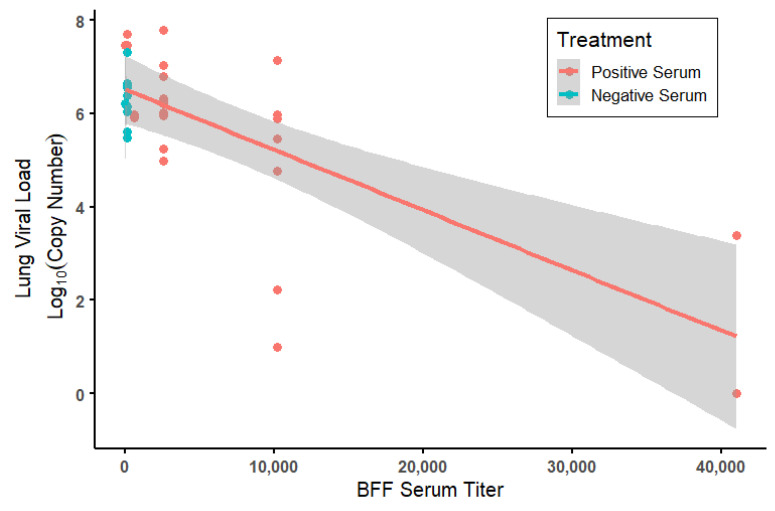
Log_10_ viral loads found in the right lungs of SARS-CoV-2 challenged mice in relation to the titer of the black-footed ferret (BFF) serum they received prior to challenge as measured by enzyme-linked immunosorbent assays (ELISA). Each point represents an individual mouse; shading is the standard error for the linear regression represented by the lines. Treatment indicates whether the BFF serum was positive for anti-SARS-CoV-2 antibodies or negative (based on the vaccination status of donor BFF).

**Table 1 viruses-14-02188-t001:** Anti-SARS-CoV-2 antibody titers in black-footed ferrets determined by enzyme linked immunosorbent assays (ELISA) and plaque reduction neutralization assays (PRNT_50_) by treatment, age, and time post-vaccination. ELISA titers ≤ 160 are considered background levels and, thus, negative.

			ELISA Titers				PRNT_50_ Titers
Ferret	Age (Years)	Treatment	Pre-Vac	Pre-Boost	2–3 Wk Post-Boost	12 Wk Post-Boost	2–3 Wk Post-Boost
8593	4	Vaccine	NA	640	2560	640	Neg
8847	4	Vaccine	160	160	10,240	2560	Neg
8995 *	3	Vaccine	160	2560	10,240	2560	<32
9135	2	Vaccine	160	2560	10,240	640	<16
9189	2	Vaccine	160	10,240	10,240	2560	256
9219	2	Vaccine	160	10,240	10,240	10,240	<64
9337	1	Vaccine	40	10,240	10,240	2560	<64
9371	1	Vaccine	40	10,240	10,240	2560	256
9422	1	Vaccine	160	10,240	40,960	2560	256
9469	1	Vaccine	160	10,240	10,240	2560	<64
9476	1	Vaccine	160	640	10,240	2560	Neg
9568	1	Vaccine	40	2560	10,240	2560	<64
9571	1	Vaccine	40	10,240	40,960	160	<64
9576	1	Vaccine	40	10,240	40,960	160	256
9598 *	1	Vaccine	160	160	2560	40	Neg
8710 *	4	Placebo	160	160	160	160	Neg
8721	4	Placebo	160	160	160	160	Neg
8970	3	Placebo	160	160	160	160	Neg
9247	2	Placebo	160	160	160	160	Neg
9311	2	Placebo	160	160	160	160	Neg
9388	1	Placebo	40	40	40	40	Neg
9425	1	Placebo	160	160	160	160	Neg
9519	1	Placebo	40	40	40	40	Neg
9592	1	Placebo	160	160	160	40	Neg

* Individuals that tested PCR positive for ferret coronavirus in fecal samples collected.

**Table 2 viruses-14-02188-t002:** Virus shedding in six black-footed ferrets following challenge with SARS-CoV-2, two of which had been vaccinated previously against the virus. Viral loads in oropharyngeal swabs and nasal flush fluids are quantified as plaque-forming units (PFU)/mL.

Ferret	Vaccinated	PFU/mL—Oropharyngeal Swab				PFU/mL—Nasal Flush Fluid			
		Day 1	Day 3	Day 5	Day 7	Day 1	Day 3	Day 5	Day 7
8822	No	<10	<10	NA	NA	40	1600	NA	NA
8305	Yes	<10	5400	NA	NA	630	29,000	NA	NA
8847	Yes	<10	<10	<10	<10	<10	<10	<10	<10
8226	No	<10	20	<10	120	1500	18,000	1200	270
9072	No	<10	140	10	130	90	1900	<10	60
8215	No	<10	60	<10	10	60	18,000	470	280

**Table 3 viruses-14-02188-t003:** Anti-SARS-CoV-2 plaque reduction neutralization titers (PRNT_90_) in black footed ferrets challenged with SARS-CoV-2 (by vaccination status, sex, and age at challenge) and tissue burdens of virus at day 3 post-challenge in euthanized individuals in plaque-forming units (PFU/gram).

Ferret	Vaccinated	Sex	Age	PRNT_90_ Titer		PFU/Gram—Day 3 Post-Challenge		
				Day 3	Day 14	Turbinates	Cranial Lung	Caudal Lung
8822	No	M	5	<10	NA	2.8 × 10^4^	<100	<100
8305	Yes	F	7	<10	NA	4.8 × 10^7^	<100	1.8 × 10^7^
8847	Yes	M	5	320	10,240	NA	NA	NA
8226	No	F	7	<10	40	NA	NA	NA
9072	No	M	4	<10	10	NA	NA	NA
8215	No	M	7	<10	40	NA	NA	NA

## Data Availability

Data are available at https://doi.org/10.5066/P9GZEXN9.

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
