# Peer review of "Immunogenicity, Safety, and Anti-Viral Efficacy of a Subunit SARS-CoV-2 Vaccine Candidate in Captive Black-Footed Ferrets (Mustela nigripes) and Their Susceptibility to Viral Challenge"

_viruses, 2022, doi:10.3390/v14102188_

Round 1

Reviewer 1 Report

The paper from Leon et al. describes black-footed ferrets’ immune response to a SARS-CoV-2 S1 subunit vaccine and the therapeutic effect of vaccinated ferrets’ serum in mice. The authors also investigated the susceptibility of BBF to SARS-CoV-2 infection, which is important to understand the SARS-CoV-2 transmission from a public health perspective. The results are interesting but there are still some points that need to be addressed.

1. line 95, the S1 subunit protein was mixed with an adjuvant as a vaccine. What strain of the SARS-CoV-2 is the S1 domain from?

2. table 1, the author suggested the youngest animals responded the greatest to vaccination. However, #9598 of 1-year old showed no antibody response at pre-boost and merely 2560 at 2-3 weeks post boost. Meanwhile, no neutralizing Ab was detected in that animal.

3. line 236, "PRNT titers only moderately correlated with ELISA titers" authors reported differences between ELISA and PRNT Abs titers. Do the results from passive serum transfer studies in mice stronger correlation with ELISA or PRNT titers? Authors should at least describe and discuss this part.

4. line 254 “3.3. Passive serum transfer studies in mice”. the design for this part is a little bit confusing to me currently. a table or figure describing the experimental challenge will make it clear to the reader.  And a summarized table will make the results more readable.

5. for the histopathology part of the whole paper (3.5 & 3.7), the authors should provide representative figures to show lesions from the groups

6. as for figure presentation, authors can add asterisks to show statistical significance if there any

Reviewer 2 Report

·       The title of the presented manuscript is (Immunogenicity, safety, and anti-viral efficacy of a subunit SARS-CoV-2 vaccine candidate in captive black-footed ferrets (Mustela nigripes) and their susceptibility to viral challenge). where is the infection susceptibility of BFF part in this study ?? or the authors meant the infection susceptibility post vaccination, or the protection capacity ?

·       S1 protein used in vaccination was used as in 50µg/dose, what is the rational of using this antigen content ?

·       What is the genetic variant (clade) of the protein used in vaccination and for the virus used in challenge infection.

·       Regarding the passive immunization of mice, what was the purification or verification method used to ensure that host-to-host transmission won’t cause any adverse reaction ? or what is the reference this research based on? .

·       what was the antibody titer of BFF serum? in figures authors mentioned the serum titer, however everywhere else in the manuscript they mentioned volume (not titer)/dose.

·       Control vaccinated mice that received monoclonal anti-S1 antibodies, what was the dosage received? And was it comparable to the serum titers?

·       Challenged mice morbidity and weight/day figures are not supplemented.

·       Why no cellular immunity measured, only humoral immunity.

·       Figure 2: was the difference in antibody titers in (ELISA) for the different age groups significant? And please explain.

·       Also in figures 3 and 4 is the difference between these groups significant ?

Reviewer 3 Report

The manuscript under consideration studied the use of COVID S1 protein as a vaccination candidate in captive black-footed ferrets. The result demonstrates that S1 subunit protein can develop protective antibodies, despite the later finding that black-footed ferrets are unsusceptible to SARS-CoV-2.

Unanticipated animal reactions have a negative impact on the robustness of mice SARS-challenging conclusions. Nonetheless, it is evident that black-footed ferrets are resistant to SARS-Cov-2.

Overall, I find the study to be well-executed, and I have no more comments. Congratulations for your excellent job.

Round 2

Reviewer 1 Report

The revised version answered my questions. Thank you.